# Outcomes of Different Haploidentical Transplantation Strategies from the Taiwan Blood and Marrow Transplantation Registry

**DOI:** 10.3390/cancers14041097

**Published:** 2022-02-21

**Authors:** Xavier Cheng-Hong Tsai, Tzu-Ting Chen, Jyh-Pyng Gau, Po-Nan Wang, Yi-Chang Liu, Ming-Yu Lien, Chi-Cheng Li, Ming Yao, Bor-Sheng Ko

**Affiliations:** 1Division of Hematology, Department of Internal Medicine, National Taiwan University Hospital, Taipei 100225, Taiwan; xavierchtsai@ntu.edu.tw (X.C.-H.T.); yaomingmd@ntu.edu.tw (M.Y.); 2Genome and Systems Biology Degree Program, National Taiwan University, Taipei 10617, Taiwan; 3Department of Hematological Oncology, National Taiwan University Cancer Center, Taipei 106037, Taiwan; 4Division of Hematology and Oncology, Department of Medicine, China Medical University Hospital, Taichung 404332, Taiwan; little_zi2000@yahoo.com.tw (T.-T.C.); leinmirain@hotmail.com (M.-Y.L.); 5Division of Hematology, Department of Medicine, Taipei Veterans General Hospital, Taipei 112201, Taiwan; jpgau@vghtpe.gov.tw; 6Division of Hematology, Department of Internal Medicine, Chang Gung Medical Foundation, Linkou Branch, Taoyuan 333423, Taiwan; dwang1415@gmail.com; 7Division of Hematology-Oncology, Department of Internal Medicine, Kaohsiung Medical University Hospital, Kaohsiung 807377, Taiwan; ycliu@kmu.edu.tw; 8College of Medicine, Kaohsiung Medical University, Kaohsiung 807378, Taiwan; 9Tai-Cheng Cell Therapy Center, National Taiwan University, Taipei 106037, Taiwan; kevinlcc1234@gmail.com; 10Department of Hematology and Oncology, Buddhist Tzu-Chi General Hospital, Hualien 970473, Taiwan

**Keywords:** haploidentical, bone marrow stem cells, peripheral blood stem cells, GIAC, PTCy

## Abstract

**Simple Summary:**

Haploidentical hematopoietic stem cell transplantation haplo-HSCT is now increasingly recognized as a valid treatment for patients with hematologic malignancies. The two most noteworthy strategies are posttransplantation cyclophosphamide (PTCy) with or without anti-thymoglobulin and granulocyte colony stimulating factor-primed bone marrow plus peripheral blood stem cells (GIAC). Direct comparisons of these approaches are rare, which makes physicians hard to choose the optimal treatment strategy for patients. We used a nationwide blood and marrow transplantation registry to compare these approaches. We found that patients in the modified GIAC (mGIAC) group had the most favorable platelet and neutrophil engraftment kinetics but had a higher extensive chronic graft-versus-host disease rate. The patients receiving mGIAC had the lowest nonrelapse mortality and highest overall survival rates. Physicians can choose the optimal treatment for patients based on the distinct clinical features and outcomes of these strategies. This study may pave the way for further prospective trials.

**Abstract:**

Background: The two most noteworthy strategies for haploidentical stem cell transplantation (haplo-HSCT) are posttransplantation cyclophosphamide (PTCy) with or without thymoglobulin (ATG) and granulocyte colony stimulating factor-primed bone marrow plus peripheral blood stem cells (GIAC). We aimed to compare these approaches in patients with hematological malignancies. Methods: We enrolled 178 patients undergoing haplo-HSCT, including modified GIAC (mGIAC), PTCy without ATG, and PTCy with ATG. Results: The patients in the mGIAC group had the most favorable platelet and neutrophil engraftment kinetics. Although the grade III–IV acute graft-versus-host-disease (GvHD) rates were similar, those receiving mGIAC had a significantly higher extensive chronic GvHD rate. The patients receiving mGIAC had a similar cumulative incidence of relapse (CIR) to that in the patients receiving PTCy with ATG, but this was lower than that in the patients receiving PTCy without ATG. The patients receiving mGIAC had the lowest nonrelapse mortality (NRM) and the highest overall survival (OS) rates. The differences in CIR, NRM, and OS remained significant when focusing on patients with low/intermediate-risk diseases before haplo-HSCT. Intriguingly, among patients with high/very-high-risk diseases before haplo-HSCT, no differences were observed in the CIR, NRM, OS, or GvHD/relapse-free survival. Conclusion: the mGIAC approach may yield a better outcome in Taiwanese patients with hematologic malignancies, especially for those with low/intermediate-risk diseases.

## 1. Introduction

Allogeneic hematopoietic stem cell transplantation (allo-HSCT) is a potential curative treatment for hematologic malignancies. However, due to the limited availability of matched related [1] or unrelated donors [2], haploidentical related donors are now increasingly considered a valid option. Regardless, bidirectional reactions to mismatched human leukocyte antigen (HLA) molecules in haploidentical hematopoietic stem cell transplantation (haplo-HSCT) are associated with a high risk of adverse immune reactions driven either by donor alloreactive T cells against recipient tissues, i.e., graft-versus-host disease (GvHD), or by host alloreactive T cells against the graft, i.e., graft rejection.

The first safe and effective approach to haplo-HSCT was a T-cell-depleted protocol consisting of a myeloablative and immunosuppressive conditioning regimen; however, this approach is associated with a high infection-related mortality rate due to slow donor posttransplant immune reconstitution [3].

A breakthrough in the field has been the development of feasible and effective T-cell-replete haplo-HSCT protocols based on novel strategies to control T-cell alloreactivity and induce T-cell tolerance. One main protocol is the administration of high-dose cyclophosphamide following graft infusion, termed the posttransplantation cyclophosphamide (PTCy) approach, which allows the immunomodulation of alloreactivities but spares donor hematopoietic stem cells [4]. This approach has become widely adopted in recent years and has evolved from using unmanipulated bone marrow (BM) to peripheral blood (PB) stem cell grafts [5,6] either combined with anti-thymocyte globulin (ATG) [7,8] or not [9,10]. The other T-cell-replete haplo-HSCT protocol consists of four components based on T-cell immunity modulation: ‘G’-CSF mobilization, ‘I’ntensified posttransplantation immunosuppression, ‘A’TG to prevent GvHD and aid engraftment, and the ‘C’ombination of bone marrow and peripheral blood stem cell grafts, which is known as the GIAC approach [11,12].

Although the outcomes in patients receiving haplo-HSCT, regardless of whether they receive the PTCy or GIAC approach, have been reported to be comparable to those in patients undergoing HLA-matched-related or HLA-matched-unrelated allo-HSCT [13,14,15], direct comparisons of these approaches are rare. However, in Taiwan, both the PTCy and GIAC haplo-HSCT protocols are used in clinical practice, and their outcomes can be observed simultaneously in an Asian real-world setting. In this study, we therefore aimed to compare these two approaches in patients with hematological malignancies in terms of the graft kinetics, toxicities, incidence of GvHD, and survival using data from the Taiwan Blood and Marrow Transplantation Registry (TBMTR).

## 2. Methods and Materials

### 2.1. Patients and Stem Cell Source

This was a retrospective, observational study based on a review of the data in the TBMTR. The TBMTR is operated by the Taiwan Society of Blood and Marrow Transplantation (TBMT) and is tasked with collecting clinical data regarding hematopoietic stem cell transplantation from all 17 Taiwan collaborative transplantation centers. Data quality control with regard to accuracy and consistency was conducted internally in the TBMTR. The data collection and analysis were approved by the institutional review board of each participating hospital, and the collection proceeded after written informed consent was obtained in accordance with the principles of the Declaration of Helsinki. 

We enrolled all the patients with hematologic malignancies who received haplo-HSCT and were registered in the TBMT database from 2011 to 2019, including those who received the modified GIAC approach (denoted as “mGIAC,” because of the inability to source M–CCNU in Taiwan) or PTCy with or without ATG. Haplo-HSCT was defined as the receipt of stem cells from a parent, offspring, or sibling with at least two or more mismatched HLA loci [13,16]. The disease risk group before haplo-HSCT was categorized according to the refined Disease Risk Index (DRI) [17]. All patients in the mGIAC group received BM stem cells on day 0 and PB stem cells on day +1, while all patients in the PTCy group either with or without ATG received purely PB stem cells on day 0.

### 2.2. Conditioning Regimens and Graft-Versus-Host-Disease Prophylaxis/Diagnosis

Myeloablative conditioning (MAC) was defined as a regimen containing either total body irradiation (TBI) with a dose equal to or greater than 8 Gy or a total dose of intravenous busulfan greater than 6.4 mg/kg [18]. All other regimens were defined as reduced intensity conditioning (RIC). The MAC in the mGIAC protocol was composed of fludarabine 30 mg/m^2^ from days −8 to −6, busulfan 3.2 mg/kg/day or TBI 1.5 Gy/fraction twice daily from days −8 to −5, and cyclophosphamide 60 mg/kg/day from days −4 to −3. The immunosuppressive agents included rabbit ATG (Thymoglobuline^®^, Genzyme, Lyon, France) 3 mg/kg/day on days −2 and −1, along with standard methotrexate (15 mg/m^2^/day on day +2 and 10 mg/m^2^/day on days +4, +7, and +12) and cyclosporine (3 mg/kg/day from day −1). The RIC in the mGIAC protocol was composed of fludarabine 30 mg/m^2^ from days −8 to −4, busulfan 3.2 mg/kg/day on days −7 and −6, and cyclophosphamide 60 mg/kg on day −3. The rabbit ATG dose was the same as that in the MAC in the mGIAC protocol. Other immunosuppressants included standard cyclosporine (3 mg/kg/day from day −1) and mycophenolate mofetil (360 mg twice daily from D + 1) (Appendix A). 

The PTCy protocol followed the original regimen, consisting of a RIC or MAC preparation regimen with cyclophosphamide 50 mg/kg/day on days +3 and +4 with mesna [4,5,19]. Other immunosuppressants included cyclosporine (3 mg/kg/day from day +5) and mycophenolate mofetil (15 mg/kg three times daily from day +5). In the PTCy with ATG group, rabbit ATG was prescribed (2.0 mg/kg/day on days −2 and −1). The diagnosis and grading of acute and chronic GVHD were based on the standard criteria [20] and revised Seattle criteria [21].

### 2.3. Endpoint Definitions and Statistical Analysis

Neutrophil engraftment was defined as the first day on which an absolute neutrophil count greater than or equal to 0.5 × 10^9^/L was achieved that lasted for three consecutive days. Platelet engraftment was defined as the first of seven consecutive days with a count greater than or equal to 20 × 10^9^/L without transfusion or other supports. OS was defined as the time from allo-HSCT to death regardless of the cause. GvHD/relapse-free survival (GRFS) was defined as the time from allo-HSCT to either the development of grade III–IV acute GvHD (aGvHD) or extensive chronic GvHD (cGvHD) or disease relapse or death. Nonrelapse mortality (NRM) was defined as death from any cause other than relapse. The cumulative incidence of relapse (CIR) was defined as recurrent disease after HSCT. For patients receiving HSCT without prior remission, the presence of disease after hemogram recovery was considered an event [22].

Continuous and discrete variables were compared using Kruskal–Wallis and chi-square tests, respectively. We used Kaplan–Meier analysis to plot the survival curves, log-rank tests to evaluate the statistical significance, and the Cox proportional hazards model for multivariate regression analysis. Cumulative incidence functions were used to estimate the neutrophil and platelet engraftment, aGvHD, cGvHD, CIR, and NRM. All models were checked to satisfy assumptions, and all statistical analyses were performed with IBM SPSS Statistics v23 (IBM Corp., Armonk, NY, USA) and R software (https://cran.r-project.org/, accessed on 18 June 2021).

## 3. Results

### 3.1. Basic Characteristics

The patient, disease, transplant, and donor characteristics of this cohort are summarized in Table 1. A total of 178 patients were enrolled, including 110 treated with mGIAC, 26 with PTCy without ATG, and 42 with PTCy with ATG. In our cohort, patients receiving mGIAC were significantly younger than those receiving PTCy with or without ATG (*p* = 0.031). Except for non-Hodgkin lymphoma and myelodysplastic syndrome/myeloproliferative neoplasm, the disease subtypes were comparable among these groups. All allografts in the PTCy strategy were from PB stem cells. There were no differences in terms of pre-HSCT disease status, conditioning intensities, donor-recipient sex combinations, or donor-recipient CMV serostatus among these groups. Of note, approximately half of the patients in this study had a high/very-high-risk diseases before allo-HSCT. There were more offspring donors in the PTCy groups with or without ATG than in the mGIAC group (*p* = 0.044). Patients in the PTCy group received more CD34+ progenitor cells than those in the mGIAC group (*p* < 0.001, Appendix A). The patients receiving PTCy either with or without ATG were spread out through the 17 transplantation centers, but patients with mGIAC, who require a bone marrow harvest process, were limited to three medical centers. There was no significant outcome difference among the centers. The median follow-up duration was 32.0 (range, 0.3–90.6) months for the entire cohort, 41.5 (range, 0.3–90.6) months for the mGIAC group, 32.0 (range, 0.3–34.9) months for the PTCy without ATG group, and 26.8 (range, 1.0–73.1) months for the PTCy with ATG group.

### 3.2. Engraftment Kinetics and Graft–Versus–Host Disease

The three strategies yielded similar 60–day neutrophil engraftment rates (99.3% for mGIAC, 97.6% for PTCy with ATG, and 92.3% for PTCy without ATG, *p* = 0.113) but distinct 100–day platelet engraftment rates (94.2% for mGIAC, 90.5% for PTCy with ATG, and 68.2% for PTCy without ATG, *p* = 0.001). Furthermore, the neutrophil engraftment times were significantly different among these three groups (median 12 days for mGIAC vs. 15 days for PTCy with ATG vs. 17 days for PTCy without ATG, *p* < 0.001) (Figure 1A). Similarly, patients in the mGIAC group also had a more favorable platelet engraftment time (median 18 days) than patients receiving PTCy with or without ATG (median 25 and 32 days, respectively) (*p* = 0.002) (Figure 1B).

The cumulative incidences of grade III–IV acute GvHD rate at 100 days were similar among the different strategies (16.4% for mGIAC vs. 14.3% for PTCy with ATG vs. 11.5% for PTCy without ATG, *p* = 0.728) (Figure 2A); however, mGIAC led to a significantly higher chronic GvHD rate than did PTCy with ATG (*p* = 0.020) (Figure 2B). The cumulative incidences of extensive chronic GvHD at 2 years were 38.8% in the mGIAC group, 8.7% in the PTCy with ATG group, and 18.6% in the PTCy without ATG group.

### 3.3. Relapse and Survival Analyses of Different Haplo–HSCT Strategies

The median time to relapse was 4.4 (range, 0.5–29.8) months for relapsed patients. The 1-year CIR in patients receiving mGIAC was significantly lower than that in patients receiving PTCy without ATG (34.5% vs. 56.1%, *p* = 0.013) but similar to that in patients receiving PTCy with ATG (38.5%, *p* = 0.861) (Figure 3A). Compared with PTCy with or without ATG, mGIAC yielded the most favorable outcome in terms of the 1-year NRM (18.5% for mGIAC, 30.5% for PTCy with ATG, and 39.1% for PTCy without ATG, *p* = 0.024) (Figure 3B) and 2-year OS (48.9% for mGIAC, 38.1% for PTCy with ATG, and 22.0% for PTCy without ATG, *p* < 0.001) (Figure 3C). However, 1-year GRFS favored PTCy with ATG (36.0%) or GIAC (33.1%) over PTCy without ATG (17.3%) (*p* = 0.032) (Figure 3D). We further performed propensity score matching analyses. With balanced baseline characteristics (Appendix A), mGIAC provided the most favorable 2-year OS (48.6%) compared with PTCy with ATG (37.9%) and PTCy without ATG (17.5%) (*p* = 0.035, Appendix A).

In multivariate Cox proportional hazards regression analysis (Table 2), after incorporating the variables with significant prognostic impact (Appendix A), pre–HSCT refined DRI was an independent predictor of CIR, GRFS, and OS. Grade III–IV acute GvHD significantly compromised NRM and GRFS; in contrast, extensive chronic GvHD was associated with better NRM and OS but also compromised GFRS. Regarding the different haplo–HSCT strategies, the use of mGIAC was an independent factor associated with better NRM and OS compared with PTCy without ATG; however, there were no significant differences when comparing with PTCy with ATG.

Regarding the causes of death (Appendix A), the majority of patients died of disease relapse, followed by infection (22.2% in the mGIAC group, 25% in the PTCy without ATG group, and 34.8% in the PTCy with ATG group) and GvHD (9.3% in the mGIAC group, 10% in the PTCy without ATG group, and 4.3% in the PTCy with ATG group).

### 3.4. Outcome Analysis of Different Haplo–HSCT Strategies Stratified by Pre–HSCT Disease Status

As half of the patients in each group had high/very-high-risk diseases before haplo–HSCT in this study, we further analyzed the outcomes stratified by disease status. For patients with low/intermediate-risk diseases, mGIAC yielded a more favorable 1-year NRM and CIR (14.8% vs. 31.7% and 49.4%, *p* = 0.011 and 15.9% vs. 28.9% and 35.2%, *p* = 0.017, respectively) than PTCy with or without ATG (Appendix A). The patients in the mGIAC group had a higher 2-year OS (72.7%) than those in the PTCy with ATG and without ATG groups (51.1% and 22.7%) (*p* < 0.001) (Figure 4A). However, GRFS was comparable among these groups (Figure 4B). Regarding the patients with high/very-high-risk diseases before haplo–HSCT, there were no significant differences among these three approaches in terms of CIR (*p* = 0.657) (Appendix A), NRM (*p* = 0.666) (Appendix A), OS (*p* = 0.191) (Figure 4C), and GRFS (*p* = 0.331) (Figure 4D).

## 4. Discussion

In recent years, haplo–HSCT has become the most common form of allo–HSCT for patients without matched donors worldwide [23,24]. Various modifications of regimens involving PTCy, including the immunosuppressants used, have been comprehensively compared [25,26,27]. Nevertheless, there had been no comparison between PTCy and the other widely used approach, GIAC. This is the first study to compare the GIAC and PTCy approaches in the same place based on real–world registry data. As M–CCNU is not available in Taiwan, we modified the GIAC protocol, and the outcomes in the patients receiving mGIAC in this study were comparable to those in the original studies [11,28]. All patients receiving PTCy used PB instead of BM stem cells, and it has been previously reported that, in the PTCy setting, PB and BM allografts yielded indistinguishable outcomes [6,29]. In this study, the patients receiving PTCy had similar outcomes to those in previous reports in terms of CIR [4], GvHD [4], NRM [30,31], and OS [4,30]. Of note, nearly half of the patients in each group enrolled in this study had high/very-high-risk diseases, which substantially compromised their prognosis.

We showed that mGIAC had advantages in terms of NRM and OS. This finding might be because the patients in the PTCy with or without ATG groups were older than those in the mGIAC group. Similarly, the higher proportion of patients with high/very-high-risk diseases in the PTCy without ATG group led to a higher CIR, although the small number of patients in this group makes it difficult to make any conclusions. After incorporating age, pre–haplo–HSCT disease status, conditioning intensity, and GvHD into multivariate analysis, these three approaches did not differ with regard to CIR and GRFS. The only differences were in NRM and OS between patients receiving PTCy without ATG and those receiving mGIAC. By analyzing the pre–HSCT disease status, we further showed that these survival differences among the three approaches were mainly among the patients with low/intermediate-risk diseases and not those with high/very-high-risk diseases. Another significant difference was observed in the engraftment kinetics. In addition to the higher neutrophil and platelet engraftment rates, the mGIAC approach yielded a significantly faster time to engraftment than PTCy with or without ATG. These findings confirmed previous descriptions of haplo–HSCT engraftment in different studies [4,11,32].

ATG has been shown to effectively prevent GvHD, both acute [33,34,35] and chronic [33,36], and it was reported that using BM stem cells might lead to a lower GvHD rate than using PB stem cells [29,37,38]. When we focused on the PTCy without ATG and mGIAC groups in which BM was used to be combined with PB as the stem cell source and ATG was used for GvHD prophylaxis, the patients receiving PTCy without ATG had a trend towards much lower grade III–IV acute GvHD and chronic GvHD, and the statistical nonsignificance might be due to the limited patient number in the PTCy without ATG group (Figure 2A,B). The differences in the rates of GvHD were more distinct when comparing the PTCy with ATG and mGIAC groups. These findings suggest that PTCy is a potent immunosuppressive strategy that could provide more favorable GvHD profiles compared with introducing BM stem cells. 

An important issue after haplo–HSCT is cytomegalovirus (CMV) infection. Due to the retrospective and registry–based nature of the study, we did not analyze the incidence of CMV infections or the associated characteristics given that there are presumed to be potential biases in the diagnosis and treatment of CMV infections. Letermovir, a CMV terminase inhibitor, was demonstrated to effectively reduce the incidence of CMV infections in patients receiving allo–HSCT in a phase 3 trial [39]. Reimbursement for letermovir by the National Health Insurance of Taiwan started in June 2020 for patients receiving haplo–HSCT. At the National Taiwan University Hospital, from 2012 to June 2020, 16 out of 108 (14.8%) patients receiving the mGIAC protocol contracted CMV infections. Intriguingly, from July 2020 to April 2021, 24 patients received the mGIAC protocol with letermovir prophylaxis, and none of them developed CMV infections (*p* = 0.044). With the introduction of letermovir, the negative outcomes related to CMV infection might be substantially reduced. 

One of the limitations of this study is the lack of information on measurable/minimal residual disease before allo–HSCT in patients with acute leukemia and information on comorbidity index values. Another limitation is that the choice of the specific transplantation strategy is based on shared decision making but not on a randomization process. In order to minimize the effect of potential selection bias, we added the unbalanced factors into the multivariate analysis. 

The patient numbers were not balanced among the three groups, which might have compromised the following statistical analyses. Despite the heterogeneity in these factors, this study still provides evidence supporting that the mGIAC approach generally yielded a better outcome than PTCy without ATG in the Taiwan population; however, adding ATG to the PTCy approach could potentially ameliorate the disadvantages. 

## 5. Conclusions

The mGIAC approach may be a preferential choice for patients with low/intermediate-risk diseases in the view of NRM, CIR, or OS. Considering the retrospective and registry–based nature of this study, further prospective trials are warranted to validate these findings.

## Figures and Tables

**Figure 1 cancers-14-01097-f001:**
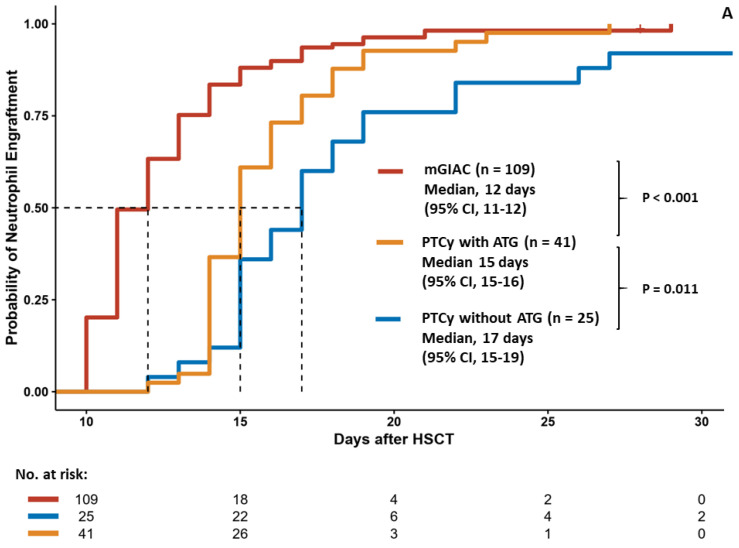
Engraftment kinetics of neutrophils (**A**) and platelets (**B**) among different haplo–HSCT approaches.

**Figure 2 cancers-14-01097-f002:**
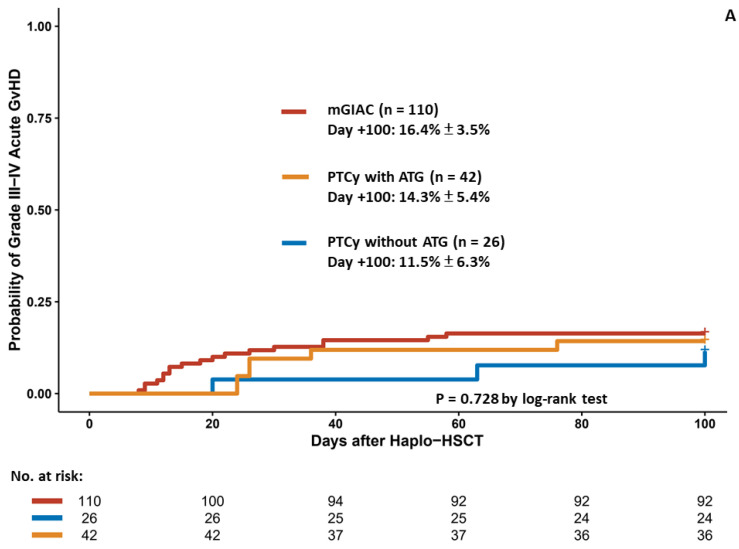
Comparison of patients receiving different haplo–HSCT approaches in terms of grade III–IV acute GvHD (**A**) and extensive chronic GvHD (**B**).

**Figure 3 cancers-14-01097-f003:**
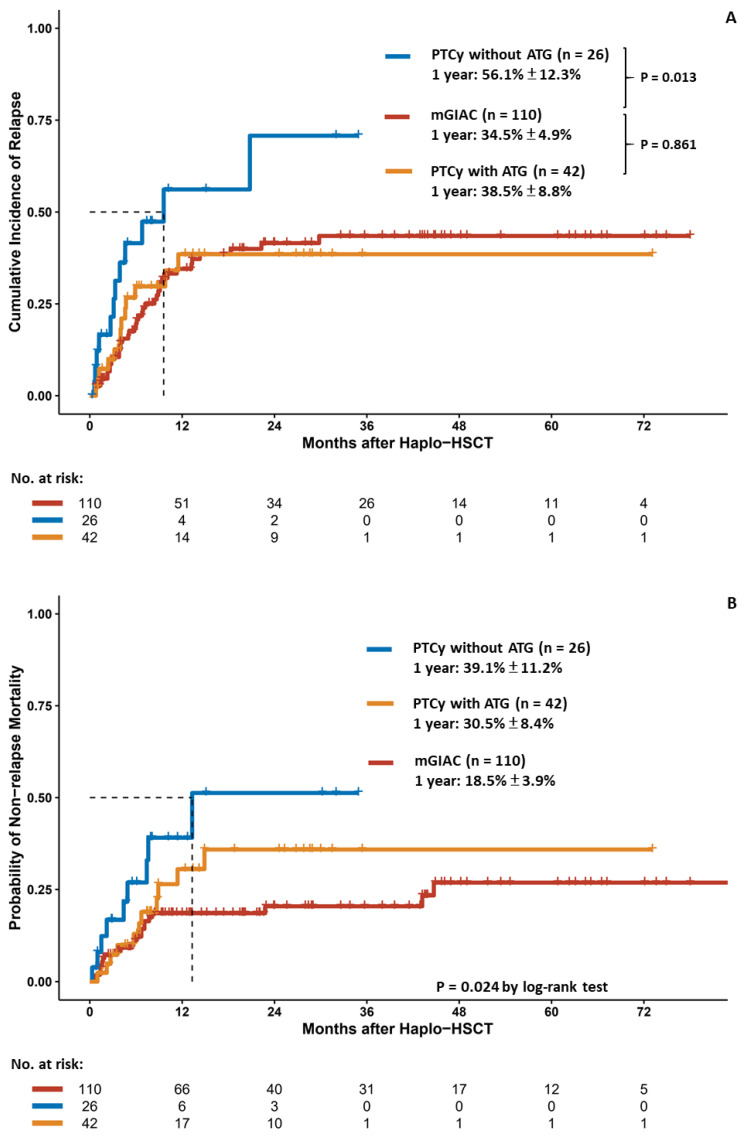
The outcome analyses of patients receiving haplo–HSCT, including the cumulative incidence of relapse (**A**), nonrelapse mortality (**B**), overall survival (**C**), and GvHD/relapse–free survival (**D**).

**Figure 4 cancers-14-01097-f004:**
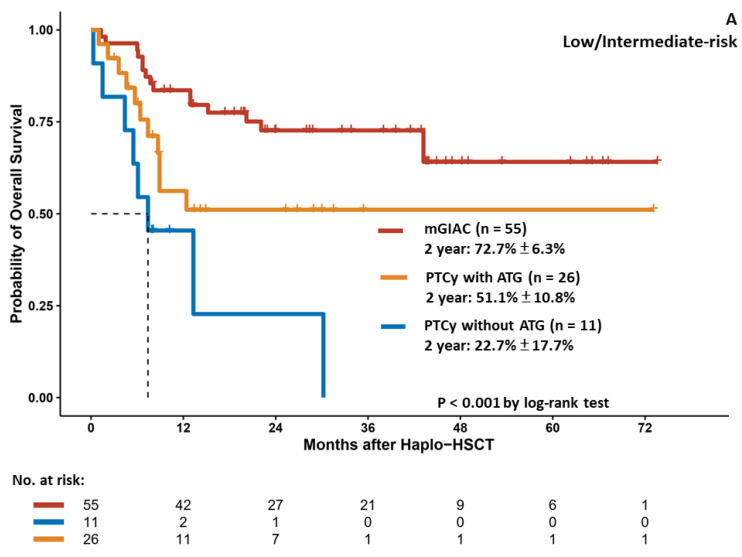
Overall survival (**A**,**C**) and GvHD/relapse–free survival (**B**,**D**) in patients with low/intermediate-risk or high/very-high-risk diseases receiving haplo–HSCT.

**Table 1 cancers-14-01097-t001:** Comparison of the clinical characteristics among patients with different haplo-HSCT approaches.

Variables	Total (n = 178)	Modified GIAC (n = 110, 61.8%)	PTCy without ATG (n = 26, 14.6%)	PTCy with ATG (n = 42, 23.6%)	*p* Value
**Sex** ^α^					0.518
Male	88 (49.4%)	52 (47.3%)	12 (46.2%)	24 (57.1%)	
Female	90 (50.6%)	58 (52.7%)	14 (53.8%)	18 (42.9%)	
**Age, years** ^βγ^	45.2 (18.7–75.6)	42.3 (18.7–69.2)	50.1 (21.8–75.6)	49.4 (18.9–68.3)	0.098
**Disease** ^α^					
AML	106 (59.6%)	65 (59.1%)	13 (50.0%)	28 (66.7%)	0.391
MDS	11 (6.2%)	9 (8.2%)	0 (0%)	2 (4.8%)	0.270
MDS/MPN	5 (2.8%)	1 (0.9%)	3 (11.5%)	1 (2.4%)	0.013
ALL	32 (18.0%)	24 (21.8%)	3 (11.5%)	5 (11.9%)	0.237
MPAL	2 (1.1%)	2 (1.8%)	0 (0%)	0 (0%)	0.535
CML	6 (3.4%)	3 (2.5%)	1 (3.8%)	2 (4.8%)	0.816
NHL	12 (6.7%)	3 (2.7%)	6 (23.1%)	3 (7.1%)	0.001
HL	3 (1.7%)	2 (1.8%)	0 (0%)	1 (2.4%)	0.748
Myeloma	1 (0.6%)	1 (0.9%)	0 (0%)	0 (0%)	0.733
**Conditioning** ^α^					0.229
Myeloablative	53 (29.8%)	25 (22.7%)	10 (38.5%)	18 (42.9%)	
Reduced intensity	125 (70.2%)	85 (77.3%)	16 (61.5%)	24 (57.1%)	
**ATG dose per kilogram** ^β^	6.0 (2.0–7.5)	6.0 (5.0–7.5)	0	4.0 (2.0–7.5)	<0.001
**Stem cell source** ^α^					<0.001
BM + mobilized PB	110 (61.8%)	110 (100%)	0 (0%)	0 (0%)	
Mobilized PB	68 (38.2%)	0 (0%)	26 (100%)	42 (100%)	
**Donor relationship** ^α^					0.106
Child	85 (47.8%)	46 (41.8%)	17 (65.4%)	22 (52.4%)	
Parent	43 (24.2%)	33 (30.0%)	2 (7.7%)	8 (19.0%)	
Sibling	50 (28.1%)	31 (28.2%)	7 (26.9%)	12 (28.6%)	
**Donor–recipient sex combination** ^α^					0.659
Female donor to male recipient	47 (26.4%)	30 (27.3%)	5 (19.2%)	12 (28.6%)	
Other combinations	131 (73.6%)	80 (72.7%)	21 (80.8%)	30 (71.4%)	
**Donor–recipient CMV serostatus** ^α^^δ^					0.073 ^γ^
Negative–Negative	3 (1.7%)	1 (0.9%)	1 (3.8%)	1 (2.4%)	
Negative–Positive	40 (22.5%)	23 (20.9%)	11 (42.3%)	6 (14.3%)	
Positive–Negative	11 (6.2%)	9 (8.2%)	1 (3.8%)	1 (2.4%)	
Positive–Positive	121 (68.0%)	77 (70.0%)	12 (46.2%)	32 (76.2%)	
Missing	3 (1.7%)	0 (0%)	1 (3.8%)	2 (4.8)	
**CD34 (10^6^/kg)** ^β^^ε^	5.08 (1.3–21.2)	5.0 (2.2–8.5)	5.87 (3.0–20.7)	6.0 (1.3–21.2)	<0.001
**Disease Risk Index** ^α^					0.069
Low	11 (6.2%)	8 (7.3%)	0 (0%)	3 (7.1%)	
Intermediate	81 (45.5%)	47 (42.7%)	11 (42.3%)	23 (54.8%)	
High	71 (39.9%)	46 (41.8%)	15 (57.7%)	10 (23.8%)	
Very high	15 (8.4%)	9 (8.2%)	0 (0%)	6 (14.3%)	
**Year of HSCT**	2016 (2011–2019)	2016 (2012–2019)	2016 (2014–2019)	2016 (2011–2019)	0.980

Abbreviations: ALL, acute lymphoblastic leukemia; AML, acute myeloid leukemia; ATG, anti–thymocyte globulin; BM, bone marrow; CMV, cytomegalovirus; CR, complete remission; HL, Hodgkin lymphoma; MDS, myelodysplastic syndrome; MPAL, mixed phenotypic acute leukemia; NHL, non–Hodgkin lymphoma; and PB, peripheral blood. ^α^ Number of patients (%). ^β^ Median (range). ^γ^ Post hoc analysis in Appendix A. ^δ^ Based on patients with available data. ^ε^ Combination of bone marrow and peripheral stem cell doses.

**Table 2 cancers-14-01097-t002:** Multivariate Cox proportional hazards regression analyses of patients receiving different haplo-HSCT strategies.

Variables	Cumulative Incidence of Relapse				Nonrelapse Mortality				GvHD/Relapse-Free Survival				Overall Survival			
	HR	Lower	Upper	*p* Value	HR	Lower	Upper	*p* Value	HR	Lower	Upper	*p* Value	HR	Lower	Upper	*p* Value
Age ^α^	0.986	0.969	1.001	0.125	1.015	0.992	1.039	0.198	0.997	0.985	1.010	0.650	1.002	0.987	1.016	0.803
Disease risk index ^β^	4.421	2..422	8.070	<0.001	1.157	0.595	2.248	0.668	1.976	1.326	2.944	0.001	2.565	1.625	4.049	<0.001
Conditioning intensity ^γ^	0.591	0.325	1.077	0.086	1.607	0.716	3.608	0.250	0.871	0.558	1.359	0.543	1.059	0.635	1.765	0.826
Acute GvHD, gr III–IV	1.460	0.748	2.850	0.267	2.431	1.124	5.260	0.024	11.327	6.485	19.785	<0.001	1.695	0.999	2.877	0.051
Chronic GvHD, extensive	0.585	0.310	1.105	0.099	0.192	0.057	0.652	0.008	1.758	1.137	2.719	0.011	0.348	0.191	0.633	0.001
Recipient CMV serostatus ^δ^	1.969	0.589	6.579	0.271	2.375	0.310	18.212	0.405	1.313	0.657	2.627	0.441	3.723	0.894	15.498	0.071
Haplo-HSCT strategies																
PTCy with ATG vs. mGIAC	1.069	0.547	2.064	0.858	1.305	0.608	2.801	0.494	1.013	0.630	1.630	0.957	1.316	0.783	2.121	0.300
PTCy without ATG vs. mGIAC	1.786	0.917	3.477	0.088	2.520	1.089	5.831	0.031	1.586	0.955	2.634	0.075	2.521	1.466	4.336	0.001

Abbreviations: ATG, anti-thymocyte globulin; and GvHD, graft-versus-host disease. ^α^ Continuous variables. ^β^ high/very-high-risk vs. low/intermediate-risk (reference). ^γ^ Reduced intensity conditioning vs. myeloablative (reference). ^δ^ Positive vs. negative (reference).

## Data Availability

The dataset analyzed in the current study cannot be publicly available because they contain historical patient data, and disclosure is prohibited by the law of privacy.

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
