# Peer review of "Outcomes of Different Haploidentical Transplantation Strategies from the Taiwan Blood and Marrow Transplantation Registry"

_cancers, 2022, doi:10.3390/cancers14041097_

Round 1
Reviewer 1 Report
The Authors have compared 3 different strategies of unmanipulated haploidentical HSCT.
posttransplantation cyclophosphamide (PTCy) with or without anti-thymoglobulin 26 (ATG) and granulocyte colony stimulating factor-primed bone marrow plus peripheral blood stem 27 cells (GIAC).
Engraftemnt was faster in the mGIAC group . The mGIAC group had a significantly higher extensive chronic GvHD rate. The patients re- 33 ceiving mGIAC had a similar cumulative incidence of relapse (CIR) to that in the patients receiving PTCy with ATG but lower than that in the patients receiving PTCy without ATG. The patients receiving mGIAC had the lowest nonrelapse mortality (NRM) and highest overall survival (OS) rates.
Interesting study on different ways of delivering haploidentical grafts.
Comments:
# the Authors should say why patients were assigned to one or the other regimen: a selection bias would be a problem for the interrpetation of these data, and this should be discussed.
# extensive chronic GvHD is a risk factor for NRM and survival: in Table 2, it reduces NRM (HR 0.18) but it increases failure in the GRFS (HR 1.69) and decreases failure in overall survival (HR 0.34) ?? something wrong!!
# in the METHODS it is not clear how GvHD prophylaxis was performed: please detail
mGIAC: ATG mg/kg , days, CSA, MTX
PTCY with ATG : ATG mg/kg days, CSA, MMF, PTCY mg/kg days
PTC without ATG : CSA MMD , PTCy mg/kg days
# the number of patients in the PTCy without ATG is small: the Authors should mention this when discussing the high reate of relapse;: it does not make sense that less GvHD prophylaxis will increase relapse. Therefore it is a question of numbers
Reviewer 2 Report
The concept of comparing mGIAC and PtCy is of importance to the haploidentical transplantation field, however, this paper has major concerns and minor concerns. All minor concerns are within the edited PDF document attached.
Major issues:
- There are no in-text references in the attached document. Please amend.
- The summary of data within the introduction and conclusion are unfounded without references.
- Despite extensive analyses, the conclusion "that haploidentical transplantation is feasible" is not at all in keeping with the data presented. If this manuscript was about feasibility there would be a focus on pragmatic aspects of these protocols and doing haplo-HSCT and the safety of regimens. This is only a minor focus of the study. Rather, a major focus is how the outcomes differ between these regimens.
- If the manuscript is about how these outcomes differ, the statistics need to be much more advanced and more appropriately presented. Namely, the impact of confounding variables of age, CD34+ dose, site of transplantation, and disease relapse on the outcomes reported. With regards to the disease relapse the authors should use disease risk index to quantify the impact of disease on outcome. With regards to more advanced statistics the authors may wish to use propensity score matching or similar techniques to allow comparison. As it stands the conclusions drawn here are not firm.
- There are multiple minor issues related to interpretation of the data that must be corrected and are discussed in the attached documents. For example considering a p vlaue of 0.541 a trend towards a difference is erroneous.

Reviewer 3 Report
This is a retrospective 'real world' study of transplant outcomes in patients undergoing haploidentical transplants using different GVHD prophylaxis and graft strategies. It is functionally a population-based analysis of the Taiwanese transplant registry. The observations are that with post transplant cyclophosphamide there is a delay in count recovery - not surprising. Overall outcomes are similar. There is a good followup time which allows for a more robust outcome analysis.
Questions on choice of transplant strategy - how was the decision made to use one of the 3 treatments? Were the patients spread out through the 14 sites? Did the PTCy pts have similar length of time in f/u?
Abstract, introduction, and methods are not clear on graft source - is bone marrow and PBSC used for both mGIAC and PTCY arms? This is discussed in the results section but would be clearer to have earlier.
Similarly define risk groups in methods
What brand(s) of rabbit ATG were used - all brands do not have the same potency.
Sorry but the difference between GIAC and mGIAC is not discussed until the discussion section (modification because of the inability to source M-CCNU). This is not commonly known in North America.
The cohort is heavily weighted to mGIAC which looks like was used in all the pediatric patients (using ALL as a surrogate) -- given the small sample size of PTCy without ATG this would bring in bias.
Table 1 -- is the CD34 calculated dose a combination of the marrow and PBSC products for the combination grafts?
Discussion:
The authors point out that the improved outcomes in mGIAC group may be due to younger population. Does the analysis hold for adult transplants (>18 y) - which would balance the numbers between the two groups. Were similar targets and duration of GVHD prophylaxis (CSA and MMF) used in both the adults and children?
Lines 265-8 are confusing. Were any of the mGIAC group bone marrow alone? - these are not reflected in the demographic table. Not clear what is being discussed
Round 2
Reviewer 2 Report
The authors have a done a commendable effort in revising the manuscript so rapidly and to address the issues highlighted. The conclusions are now adequately addressed.